# FedLite: A Scalable Approach for Federated Learning on Resource-constrained Clients

## Abstract

In classical federated learning, the clients contribute to the overall training by communicating local updates for the underlying model on their private data to a coordinating server. However, updating and communicating the entire model becomes prohibitively expensive when resource-constrained clients collectively aim to train a *large* machine learning model. Split learning provides a natural solution in such a setting, where only a (small) part of the model is stored and trained on clients while the remaining (large) part of the model only stays at the servers. Unfortunately, the model partitioning employed in split learning significantly increases the communication cost compared to the classical federated learning algorithms. This paper addresses this issue by proposing an end-to-end training framework that relies on a novel vector quantization scheme accompanied by a gradient correction method to reduce the additional communication cost associated with split learning. An extensive empirical evaluation on standard image and text benchmarks shows that the proposed method can achieve up to $490\times$ communication cost reduction with minimal drop in accuracy, and enables a desirable *performance vs. communication* trade-off.

## 1 Introduction

Federated learning (FL) is an emerging field that collaboratively trains machine learning models on decentralized data (Li et al., 2019; Kairouz et al., 2019; Wang et al., 2021). One major advantage of FL is that it does not require clients to upload their data which may contain sensitive personal information. Instead, clients separately train local models on their private datasets, and the resulting locally trained model parameters are infrequently synchronized with the help of a coordinating server (McMahan et al., 2017). While the FL framework helps alleviate data-privacy concerns for distributed training, most of existing FL algorithms critically assume that the clients have enough compute and storage resources to perform local updates on the entire machine learning model. However, this assumption does not necessarily hold in many modern applications. For example, classification problems with an extremely large number of classes (often in millions and billions) commonly arise in the context of recommender systems (Covington et al., 2016), information retrieval (Agrawal et al., 2013), and language modeling (Levy & Goldberg, 2014). Here, the classification layer of a neural network itself is large enough that a typical FL client, e.g., a mobile or IoT device, cannot even store and locally update this single layer, let alone the entire neural network.

Split learning (SL) is a recently proposed technique (Vepakomma et al., 2018; Thapa et al., 2020) that naturally addresses the above issue of FL. It splits the underlying model between the clients and server such that the first few layers are shared across the clients and the server, while the remaining layers are only stored at the server. The reduction of resource requirement at the clients is particularly pronounced when the last few dense layers constitute a large portion of the entire model. For instance, in a convolutional neural network (Krizhevsky, 2014), the last two fully connected layers take $95\%$ parameters of the entire model. In this case, if we allocate the last two layers to the server, then the client-side memory usage can be reduced by $20\times$. Nonetheless, one major limitation of SL is that the underlying model partitioning leads to an increased communication cost for the resulting framework. Specifically, to train the split neural network, the activations and gradients at the layer where the model is split (referred to as cut layer) need to be communicated between the server and

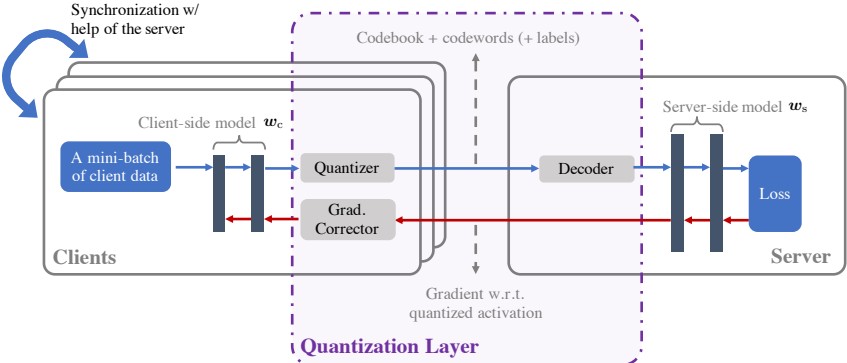

Figure 1: Overview of the proposed algorithm. In order to reduce the additional communication between the clients and the server, we propose to cluster similar client activations and send the cluster centroids to the server instead. This is equivalent to adding a vector quantization layer in split neural network training. The success of our method relies on a novel variant of product quantizer and a gradient correction technique. We refer the reader to Section 4 for further details.

clients at each iteration. The additional message size is in proportion to the mini-batch size as well as the activation size. As a result, the communication cost for the model training can become prohibitive whenever the mini-batch size and the activation size of the cut layer are large (e.g. in one of our experiments, the additional message size can be 10 times larger than that of the client-side model).

In this paper, we aim to make the split neural network training communication-efficient and enable its widespread adoption for FL in resource-constrained settings. Our proposed solution is based on the critical observation that, given a mini-batch of data, the client does not need to communicate per-example activation vectors if the activation vectors (at the cut layer) for different examples in the mini-batch exhibit enough similarity. Thus, we propose a training framework that performs clustering of the activation vectors and only communicates the cluster centroids to the server. Interestingly, this is equivalent to adding a vector quantization layer in the middle of the split neural network (see Figure 1 for an overview of our proposed method). Our main contributions are as follows.

- We propose an end-to-end communication-efficient neural network splitting-based FL approach for resource-constrained clients (cf. Section 4). The approach employs a novel compression scheme that leverages product quantization to effectively compress the activations communicated between the clients and server.

- After applying the activation quantization, the clients can only receive possibly noisy gradients from the server. The inaccurate client-side model updates lead to significant accuracy drops in our experiments. In order to mitigate this problem, we propose a gradient correction scheme for the backward pass, which plays a critical role in achieving a high compression ratio with minimal accuracy loss.

- We empirically evaluate the performance of our approach on three standard FL datasets (cf. Section 5). Remarkably, we show that our approach allows for up to $490\times$ communication reduction without significant accuracy loss.

- We present a convergence analysis for the proposed method (cf. Section 4.3), which reveals a trade-off between communication reduction and convergence speed. The analysis further helps explain why the proposed gradient correction technique is beneficial.

Here we note that our proposed method has potential applications beyond FL with resource-constrained clients as it can be applied in any learning framework that can benefit from a quantization layer. For instance, it can be used to reduce the communication overhead in two-party vertical FL (Romanini et al., 2021), where the model is naturally split across two institutions and the data labels are generated on the server. It is also worth mentioning that the proposed method does not expose any additional client-side information to the server than vanilla split neural network training approach, such as SPLITFED (Thapa et al., 2020). Thus, our method can also leverage existing privacy preserving mechanisms such as differential privacy (Wei et al., 2020) or instance-hiding schemes (Huang et al., 2020) to provide formal privacy guarantees.

## 2 BACKGROUND AND RELATED WORKS

**Federated Learning (FL).** Suppose we have a collection of $M$ clients $\mathcal{I} = \{1, 2, \ldots, M\}$. Each client $i \in \mathcal{I}$ has a local dataset $\mathbb{D}_i$ and a corresponding empirical loss function $F_i(\boldsymbol{w}) = \sum_{\xi \in \mathbb{D}_i} f(\boldsymbol{w}; \xi)/n_i$, where $\boldsymbol{w}$ denotes the model parameters, and $n_i = |\mathbb{D}_i|$ denotes the number of samples of the local dataset. The goal of FL is to find a shared model $\boldsymbol{w}$ that can minimize the averaged loss over all clients, defined as follows:

$$F(\boldsymbol{w}) = \sum_{i=1}^{M} p_i F_i(\boldsymbol{w}) \tag{1}$$

where $p_i = n_i / \sum_{i=1}^{M} n_i$ is the relative weight of local loss $F_i$. Motivated by the data privacy concerns, under a FL framework, the clients perform local training and only communicate the resulting models to a coordinating server as opposed to sharing their raw local data with the server (McMahan et al., 2017). In current FL algorithms, the model size is limited by the resource constraints of clients. Consequently, these algorithms are not feasible when dealing with large machine learning models that cannot fit in clients memory or require large compute. Compared to these vanilla FL algorithms, split learning-based approaches enables learning on resource-constrained clients at a cost of additionally exposing the label information of clients to the server.

**Large Model Training in FL.** To deploy large machine learning models on resource-constrained clients, a few recent works proposed methods to reduce the effective model size on clients. For example, Diao et al. (2020) varied the layer width on clients depending on their computational capacities; and Horvath et al. (2021) proposed ordered dropout to randomly removing neurons in the neural network. These methods are effective in reducing the width of the intermediate layers; however, they still place the full classification layer (which is in proportion to the size of the output space) on each client to compute the the client update. This may not be feasible for many clients as the parameters in the classification layer of a model can dominate the total number of model parameters (Krizhevsky, 2014).

**Split Learning (SL).** Split learning (SL) is another way of minimizing (1) without explicitly sharing local data on clients (Vepakomma et al., 2018). In particular, SL splits the neural network model into two parts on a layer basis. The first few layers (called client-side model parameterized by $\boldsymbol{w}_c$) are shared across clients and the server, while the remaining layers (called server-side model parameterized by $\boldsymbol{w}_s$) are only stored and trained on the server. Under this splitting, the original loss function for a data sample $\xi$ can be re-written as follows:

$$f(\boldsymbol{w}; \xi) = h(\boldsymbol{w}_s; u(\boldsymbol{w}_c; \xi)), \quad \forall \xi \in \mathbb{D}_i, \forall i \in \mathcal{I} \tag{2}$$

where $u$ is the client-side function mapping the input data to the activation space and $h$ is the server-side function mapping the activation to a scalar loss value. Training both the client- and server-side models requires communicating client activations (i.e., the output of the client-side model, also called smashed data in some literature) between the clients and the server. We further elaborate on a concrete SL algorithm as a baseline in Section 3.

**Communication-efficient SL and Model Parallel Training.** Recently, He et al. (2020); Han et al. (2021) proposed to add a classification layer on the client-side model in SL so that each client can locally update its parameters. This can effectively reduce the communication frequency between the clients and the server. However, this kind of method is not suitable for the settings where the classification layer dominates the entire model size such that the clients may not have enough memory space to store the entire classification layer.

Another relevant approach is reducing communication cost in model parallel (MP) training (Dean et al., 2012). In MP training the worker nodes (dedicated machines instead of resource-constrained client devices) also need to transfer activations of intermediate layers with each other. Gupta et al. (2020) proposed to sparsify the activations according to the magnitude of each element and achieved up to $20\times$ compression ratio (uncompressed/compressed). However, in practice it might be tricky to set the threshold for sparsification. Our proposed method explores an orthogonal dimension of sparsification by clustering similar activations within the mini-batches and communicating only the cluster centroids. Empirical results show that the proposed method can achieve up to $490\times$ compression ratio without a significant drop in the model performance.

**Product Quantization.** Product quantization (PQ) is a compression method and has been widely used in approximate nearest neighbor search (Jegou et al., 2010; Ge et al., 2013b). Given a batch of vectors, PQ divides each vector into multiple chunks and performs K-means clustering separately on each chunk. Now, the resulting cluster centroids construct the codebook and the closest codeword will be used to represent each vector. Instead of directly computing the distance between two high-dimensional vectors, computing the distance between two codewords is orders of magnitude more efficient and faster. Note that the previous works used PQ to compress either learned features after training (see, e.g., Ge et al., 2013a) or the parameters of a neural network (see, e.g., Chen et al., 2020). In this paper, we show that PQ is also effective in compressing the output of intermediate layers and present a simple technique to enable backpropagation through the corresponding quantization layer.

## 3 BASELINE: THE SPLITFED ALGORITHM

In this section, we review a representative FL algorithm based on split learning, namely SPLITFED (Thapa et al., 2020), that provides a baseline training approach in resource-constrained setting. Each iteration of SPLITFED contains four steps detailed as follows. Wherever it simplifies the presentation, we assume that the mini-batch size $B$ on each client is 1 as the derivations hold true for arbitrary batch sizes.

1. **Client Forward Pass**: Let $\mathcal{S}$ be a randomly selected subset of clients. For $i \in \mathcal{S}$, compute the output of the client-side model $\boldsymbol{z}_i = u(\boldsymbol{w}_c; \xi)$, where $\xi \in \mathbb{D}_i$ is a randomly chosen training sample, and $\boldsymbol{z}_i \in \mathbb{R}^d$ denotes the activations of the last layer of the client-side model. Then, each selected client sends $\boldsymbol{z}_i$ (together with its corresponding label if necessary) to the server.

2. **Server Update**: The server treats all the activations (also referred to as smashed data) $\{\boldsymbol{z}_i\}_{i \in \mathcal{S}}$ as inputs to perform one step of gradient descent on the server-side model: $\Delta \boldsymbol{w}_s = \eta_s \sum_{i \in \mathcal{S}} p_i \nabla_{\boldsymbol{w}_s} h(\boldsymbol{w}_s; \boldsymbol{z}_i) / \sum_{i \in \mathcal{S}} p_i$, where $\eta_s$ denotes the server learning rate.[1] In addition, the server also computes the gradient with respect to the activations $\nabla_{\boldsymbol{z}_i} h(\boldsymbol{w}_s; \boldsymbol{z}_i)$ and sends it back to the corresponding client $i$.

3. **Client Backward Pass**: Each selected client computes the gradient with respect to the client-side model using the chain rule: $\boldsymbol{g}_i \triangleq \nabla_{\boldsymbol{w}_c} f = \nabla_{\boldsymbol{z}_i} h(\boldsymbol{w}_s; \boldsymbol{z}_i) \nabla_{\boldsymbol{w}_c} u(\boldsymbol{w}_c; \xi)$.

4. **Client-side Model Synchronization**: At last, the client-side model is updated by synchronizing client gradients $\Delta \boldsymbol{w}_c = \eta_c \sum_{i \in \mathcal{S}} p_i \boldsymbol{g}_i / \sum_{i \in \mathcal{S}} p_i$, where $\eta_c$ denotes the learning rate for the client-side model.

One can easily validate that, with the above four steps, SPLITFED is equivalent to mini-batch stochastic gradient descent (SGD) with a total batch size of $B \times |\mathcal{S}|$. No matter how the network is split, the performance of the algorithm remains unchanged and has the same iteration complexity as mini-batch SGD.

Besides, note that, a selected client needs to upload both the activations during "Client Forward Pass" and the gradients of the client-side model during "Client-side Model Synchronization". Thus, the communication size per client per iteration can be written as $|\boldsymbol{w}_c| + Bd$. Since the up-link communication bandwidth is quite limited for clients in federated learning, when the mini-batch size $B$ or the dimension of the activation $d$ gets large, the up-link communication may become the main bottleneck that limits the scalability of SPLITFED. It is critical to compress the uploaded message from clients to the server (Reisizadeh et al., 2019; Li et al., 2019). Next, we present a novel vector quantization-based FL training framework that addresses this limitation of SPLITFED to enable communication-efficient training in resources-constrained settings.

## 4 PROPOSED METHOD: FEDLITE

We now introduce our proposed method, **FED**erated sp**LIT** learning with v**E**ctor quantization (FEDLITE), which can drastically reduce the up-link communication cost on clients in the SPLITFED algorithm. The key idea of our method is to compress the redundant information in the client activations for each mini-batch. For instance, if we assume that all the activation vectors within a

---

[1]The server learning rate here has a different definition from previous FL literature, e.g., Reddi et al. (2020).

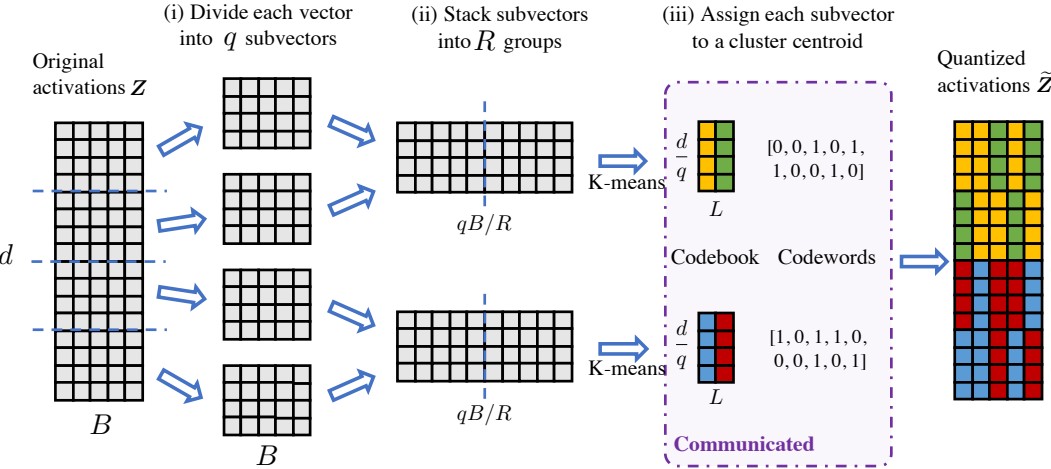

Figure 2: Illustration of our proposed quantizer. Given a mini-batch of activations $\boldsymbol{Z} \in \mathbb{R}^{d \times B}$, there are three steps: (i) divide each activation vector into $q$ subvectors; (ii) stack subvectors into $R$ groups based on their indices; (iii) perform K-means clustering to get $L$ centroids for each group. Each subvector can be represented by the index of the closest centroid in its corresponding group. On the server, by simply rearranging centroids, we can get the quantized activations $\widetilde{\boldsymbol{Z}}$.

mini-batch are nearly identical to each other, then instead of sending all $B$ activation vectors, the client needs to send only one representative vector to the server, thereby, reducing the communication cost by a factor of $B$.

Building on the above observation, we let each client input their activations into a clustering algorithm to get $L$ centroids. Then, each activation vector is represented by the index of its closest centroid. Instead of sending a mini-batch of $B$ vectors, only $L$ centroids and the cluster assignments are transmitted to the server. This procedure is equivalent to vector quantization. Here, a simple choice of the clustering algorithm is K-means (Krishna & Murty, 1999). However, as we observe in Figure 3, vanilla K-means clustering leads to a high quantization error with very limited compression ratio. It is non-trivial to design a proper clustering algorithm that can minimize the communication between clients and server, while maintaining a low quantization error. In Section 4.1, we design a novel variant of product quantizer that can flexibly operate this trade-off.

While the above approach is appealing, it also introduces new challenges. First, note that, in the backward pass, due to the additional quantization layer, the server no longer knows the true client activations; thus, constraining the server to compute the gradients with respect to the (possibly noisy) quantized activations. Furthermore, in order to update the client-side model parameters, the clients need to receive the gradients with respect to the original activations, which is not available any more. In Section 4.2, we introduce a gradient correction technique to mitigate this gradient mismatch, which is critical for our method to achieve a high compression ratio with minimal accuracy loss.

In the following sections, we present the key components of our proposed method and highlight how they address the aforementioned challenges. In addition, a convergence analysis is provided in Section 4.3. Unless otherwise stated, for the ease of exposition, we will focus on the operations on a specific client $i$ and omit the client index. However, it is important to keep in mind that the client-side operations happen in parallel on all selected clients.

## 4.1 FORWARD PASS: COMPRESSING ACTIVATIONS WITH PRODUCT QUANTIZATION

In this subsection, we first present our proposed qunatizer and then elaborate on its advantages. A visual illustration is provided in Figure 2.

**The Proposed Compression Scheme.** Suppose a client has computed a batch of activations $\boldsymbol{Z} = [\boldsymbol{z}^{(1)}, \ldots, \boldsymbol{z}^{(B)}] \in \mathbb{R}^{d \times B}$ using the client-side model $\boldsymbol{w}_{\mathrm{c}}$, where $d$ is the dimension of each activation and $B$ denotes the mini-batch size on each client. Our proposed quantizer first divides each activation vector $\boldsymbol{z}^{(j)}, j \in [1, B]$ into $q$ subvectors with equal dimension $d/q$. We

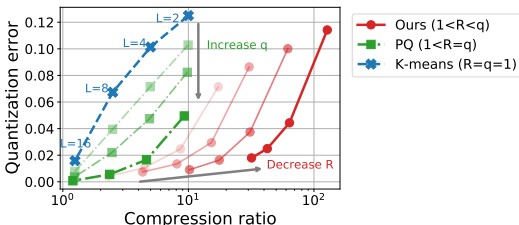

Figure 3: Quantization error (lower is better) versus compression ratio (larger is better). Our proposed quantizer achieves a better quantization error vs. compression ratio trade-off as compared to vanilla product quantization (PQ) and K-means. In this example, the activation size is $d = 9216$, and the mini-batch size is $B = 20$. The activations are trained via a two-layer CNN on Federated EMNIST. In each curve, we vary the number of clusters $L$. For green curves, the number of subvectors $q$ takes value in $\{288, 1152, 4608\}$; for red curves, the number of groups $R$ takes value in $\{2304, 1152, 384, 1\}$, and $q$ is fixed as $4608$.

denote the $s$-th subvector of $\boldsymbol{z}^{(j)}$ as $\boldsymbol{z}^{(j,s)} \in \mathbb{R}^{d/q}$. Then, the quantizer stacks all subvectors into $R$ groups based on their indices. For example, the first group $\mathcal{G}^{(1)}$ contains all subvectors with indices $(j, 1), (j, 2), \ldots (j, q/R), \forall j \in [1, B]$; the second group $\mathcal{G}^{(2)}$ contains all subvectors with indices $(j, q/R + 1), (j, q/R + 2), \ldots, (j, 2q/R), \forall j \in [1, B]$, and so forth. At last, K-means clustering is performed on subvectors within each group $\mathcal{G}^{(r)}, \forall r \in [1, R]$ to find $L$ cluster centroids $\mathcal{C}^{(r)} = \{\boldsymbol{c}^{(r,1)}, \ldots, \boldsymbol{c}^{(r,L)}\} \subset \mathbb{R}^{d/q}$. Now each subvector $\boldsymbol{z}^{(j,s)}$ can be approximated by its closest centroid in the corresponding group. Formally, if $\boldsymbol{z}^{(j,s)} \in \mathcal{G}^{(r)}$, then

$$\widetilde{\boldsymbol{z}}^{(j,s)} \triangleq \boldsymbol{c}^{(r,l_*^{(j,s)})}, \quad \text{where } l_*^{(j,s)} = \underset{l \in [1,L]}{\arg\min} \|\boldsymbol{z}^{(j,s)} - \boldsymbol{c}^{(r,l)}\|^2 \tag{3}$$

In the above equation, $\widetilde{\boldsymbol{z}}^{(j,s)}$ represents the quantized version of $\boldsymbol{z}^{(j,s)}$. By concatenating all quantized subvectors, server can get the quantized activations $\widetilde{\boldsymbol{Z}}$ and use it as input to update the server-side model. In order to obtain the quantized subvectors, each client needs to send all cluster centroids $\{\mathcal{C}^{(r)}\}_{r \in [1,R]}$ (referred to as the *codebook*) and the cluster assignments $\{l_*^{(j,s)}\}_{j \in [1,B], s \in [1,q]}$ (referred to as the *codewords*) to the server. Assuming that each floating-point number occupies $\phi$ bits[2], the transmitted message size per client is reduced to $\phi dRL/q + Bq \log_2 L$ from $\phi dB$.

**The Benefit of Subvector Division: Reduced Quantization Error.** The first key component of our proposed quantizer is subvector division, i.e., setting the number of subvectors $q$ to be greater than 1. When $q = 1$, the quantizer reduces to vanilla K-means. In this case, the codebook size is $\phi dL$ and each activation vector have $L$ possible choices. When we set $q = R > 1$, the codebook size is still $\phi dL$. However, each activation vector now has $L^q$ possible choices, as each of $q$ subvectors has $L$ choices. Thus, the number of quantization levels becomes exponentially larger without any increase in memory usage and computation complexity. As illustrated by the green lines in Figure 3, having more quantization levels or effective centroid choices can significantly lower the quantization error.

**The Benefit of Subvector Grouping: Improved Compression Ratio.** By using subvector division, although the quantization error can be reduced, the codebook size $\phi dL$ is still pretty large. This is because subvectors within one vector are quantized separately using different codebooks. To further reduce the communication size, we propose subvector grouping and force subvectors within one group to share the same codebook. As a result, the total codebook size can be reduced to $\phi dLR/q$. When $q \gg R \geq 1$, there can be an order of magnitude increase in the compression ratio, as illustrated by the red lines in Figure 3. One can also observe that, by changing the value of $R$, our proposed quantizer (red lines) achieves a much better error-versus-compression trade-off than K-means and vanilla production quantization scheme.

**Why not Reuse the Codebooks from Previous Iterations?** In our proposed scheme, the codebook is reconstructed at each iteration on clients. This is necessary, since in FL, only few stateless clients are selected to participate training in at each iteration and they have non-IID data distributions (Kairouz et al., 2019). Previous codebooks can be stale and other clients' codebooks may not be suitable.

---

[2]When computing the compression ratio in this paper, we always assume $\phi = 64$.

Now that we have introduced our product quantization-based compression scheme, we next address the issue of performing backpropagation in the presence of the additional quantization layer.

## 4.2 BACKWARD PASS: GRADIENT CORRECTION

As mentioned at the beginning of Section 4, there is a gradient mismatch problem in the client backward pass. While client $i$ needs $\nabla_{\boldsymbol{z}_i} h(\boldsymbol{w}_{\mathrm{s}}; \boldsymbol{z}_i)$ to update the client-side models, it can only receive $\nabla_{\widetilde{\boldsymbol{z}}_i} h(\boldsymbol{w}_{\mathrm{s}}; \widetilde{\boldsymbol{z}}_i)$ from the server. A naive solution is to treat $\nabla_{\widetilde{\boldsymbol{z}}_i} h$ as an approximation of $\nabla_{\boldsymbol{z}_i} h$ and update the client-side model by $\nabla_{\widetilde{\boldsymbol{z}}_i} h(\boldsymbol{w}_{\mathrm{s}}; \widetilde{\boldsymbol{z}}_i) \nabla_{\boldsymbol{w}_{\mathrm{c}}} u(\boldsymbol{w}_{\mathrm{c}}; \xi) = (\partial f / \partial \widetilde{\boldsymbol{z}}_i)(\partial \boldsymbol{z}_i / \partial \boldsymbol{w}_{\mathrm{c}})$. However, due to gradient mismatch, this approach can lead to a significant performance drop, as observed in our experiments (cf. Section 5). In order to address this issue, we propose a gradient correction technique. In particular, we approximate the gradient $\nabla_{\boldsymbol{z}_i} h(\boldsymbol{w}_{\mathrm{s}}; \boldsymbol{z}_i)$ by its first-order Taylor series expansion: $\nabla_{\widetilde{\boldsymbol{z}}_i} h(\boldsymbol{w}_{\mathrm{s}}; \widetilde{\boldsymbol{z}}_i) + \nabla^2_{\widetilde{\boldsymbol{z}}_i} h(\boldsymbol{w}_{\mathrm{s}}; \widetilde{\boldsymbol{z}}_i) \cdot (\boldsymbol{z}_i - \widetilde{\boldsymbol{z}}_i)$. While the higher-order derivative may be expensive to compute, we use a scalar parameter $\lambda \boldsymbol{I} > 0$ to replace $\nabla^2_{\widetilde{\boldsymbol{z}}_i} h(\boldsymbol{w}_{\mathrm{s}}; \widetilde{\boldsymbol{z}})_i$. Consequently, the gradient of the client-side model is defined as follows:

$$\widetilde{\boldsymbol{g}}_i \triangleq \left[ \frac{\partial h(\boldsymbol{w}_{\mathrm{s}}; \widetilde{\boldsymbol{z}}_i)}{\partial \widetilde{\boldsymbol{z}}_i} + \lambda(\boldsymbol{z}_i - \widetilde{\boldsymbol{z}}_i) \right] \frac{\partial u(\boldsymbol{w}_{\mathrm{c}}; \xi)}{\partial \boldsymbol{w}_{\mathrm{c}}}. \tag{4}$$

In Section 4.3, we will provide a convergence analysis that can help explain the effects of gradient correction; in Section 5, we will empirically show that setting a strictly positive $\lambda$ is crucial for the success of the proposed method. In the following discussion, we provide an intuitive explanation of the correction to further motivate our approach.

**The Effect of Regularization.** We show that the client-side gradient in (4) can be considered as a gradient of the following surrogate loss (proof is provided in the Appendix):

$$\|\boldsymbol{z}_i - \hat{\boldsymbol{z}}_i\|^2 + \frac{\lambda}{2} \|\boldsymbol{z}_i - \widetilde{\boldsymbol{z}}_i\|^2, \tag{5}$$

where $\boldsymbol{z}_i = u(\boldsymbol{w}_{\mathrm{c}}; \xi)$ is the activation of the client-side model. Besides, $\hat{\boldsymbol{z}}_i \triangleq \boldsymbol{z}_i - \nabla_{\widetilde{\boldsymbol{z}}_i} h(\boldsymbol{w}_{\mathrm{s}}; \widetilde{\boldsymbol{z}}_i)/2$ and the quantized activation $\widetilde{\boldsymbol{z}}_i$ are fixed vectors that do not have derivatives when computing the gradient. Setting $\lambda > 0$ is equivalent to having a regularization term, as per (5). The regularizer encourages the client-side model $\boldsymbol{w}_{\mathrm{c}}$ to move in a direction that can decrease the quantization error $\|\boldsymbol{z}_i - \widetilde{\boldsymbol{z}}_i\|$. Interestingly, using a larger value of $\lambda$ may let clients put more emphasis on minimizing the quantization error and lead to quantization-friendly activations. However, one cannot set a arbitrarily large value for $\lambda$ because the client-side model may output the same activation for all inputs and fail to minimize the original loss function.

## 4.3 CONVERGENCE ANALYSIS

In this subsection, we provide a convergence analysis for FEDLITE. The analysis will highlight how the quantization error influences the convergence and how the gradient correction technique helps. In particular, the analysis is conducted under standard assumptions for mini-batch SGD. We assume the objective function $F(\boldsymbol{w}) = F([\boldsymbol{w}_{\mathrm{c}}; \boldsymbol{w}_{\mathrm{s}}])$ is $L$-Lipschitz smooth (i.e., $\|\nabla F(\boldsymbol{w}) - \nabla F(\boldsymbol{v})\| \le L \|\boldsymbol{w} - \boldsymbol{v}\|$), and stochastic gradient $\boldsymbol{g}(\boldsymbol{w})$ has bounded variance $\mathbb{E}\|\boldsymbol{g}(\boldsymbol{w}) - \nabla F(\boldsymbol{w})\|^2 \le \sigma^2/BS$, where $B$ is the mini-batch size per client and $S$ is the number of selected clients per iteration. Under these assumptions, we have the following theorem.

**Theorem 1** (Convergence of FEDLITE)**.** *If the client-side model and server-side model update using the same learning rate* $\alpha = \sqrt{BS/T}$ *where $T$ is the number of total iterations, then the expected gradient norm of the global function* $\min_{t \in [0, T-1]} \mathbb{E} \left\| \nabla F(\boldsymbol{w}^{(t)}) \right\|^2$ *can be bounded by*

$$\underbrace{\frac{4(F(\boldsymbol{w}^{(0)}) - F_{inf})}{\sqrt{BST}} + \frac{4L\sigma^2}{\sqrt{BST}}}_{\textit{Opt. error of mini-batch SGD}} + \underbrace{\left( 4\sqrt{\frac{BS}{T}} + 2 \right) (\Lambda_1^2 + (\Lambda_2 - \lambda)^2 \Lambda_3^2) \kappa^2}_{\textit{Addnl. error caused by quantization}}. \tag{6}$$

*where $\kappa$ is the maximal quantization error* $\max \|\boldsymbol{z} - \widetilde{\boldsymbol{z}}\|$, $\lambda$ *is the tunable parameter in our gradient correction scheme, $F_{inf}$ is the lower bound of function value, and constants $\Lambda_1, \Lambda_2, \Lambda_3$ are the largest eigenvalues of matrices $\partial^2 h(\boldsymbol{w}_s; \boldsymbol{z})/\partial z \partial w_s, \partial^2 h(\boldsymbol{w}_s; \boldsymbol{z})/\partial z^2, \partial u(\boldsymbol{w}_c; \xi)/\boldsymbol{w}_c$, respectively.*

Theorem 1 guarantees that the proposed algorithm FEDLITE converges to a neighbourhood around the stationary point of the global function. And the size of this neighborhood is in proportion to the maximal quantization error $\kappa$ during the training. When there is no quantization, the error bound (6) recovers that of mini-batch SGD. Moreover, one can observe that setting a positive $\lambda$ can help to reduce the additional error caused by adding the quantization layer.

## 5 EXPERIMENTS

We implement the proposed method FEDLITE using FedJAX (Ro et al., 2021) and evaluate its effectiveness on three standard federated datasets provided by the Tensorflow Federated (TFF) package (TFF, 2021): (i) image classification on FEMNIST, (ii) tag prediction on StackOverflow (referred to as SO Tag) and (iii) next word prediction on StackOverflow (referred to as SO NWP). On FEMNIST, the same model architecture as (Reddi et al., 2020) is adopted. We place two convolutional layers and two dense layers on the clients and the server, respectively. *With this splitting, the client-side model only has about* $1.6\%$ *trainable parameters of the entire model.* Therefore, the client-side resource requirement is significantly reduced. On SO Tag, both the client- and server-side models contains only one dense layer. On SO NWP, we place one LSTM layer and one dense layer on the clients, and another dense layer on the server. The ratios between the client-side model size and entire model size are $83\%$ and $79\%$ on SO Tag and SO NWP, respectively. Here, we note that even though the client-side model sizes for SO Tag and SO NWP do not correspond to an ideal setting for the split learning-based approaches, we include these two datasets to showcase the utility of our proposed method for language-based tasks.

For each task, we select the learning rate that is best for the baseline SPLITFED algorithm. Although separately tuning the learning rate for our proposed method can further improve its performance, using the same learning rate as SPLITFED already demonstrates the advantages of our method. Unless otherwise stated, the number of groups $R$ in our proposed quantizer is set to 1, as it exhibits the best trade-off in Figure 3 and experiments.

**Main Results: Effectiveness of FedLite.** As discussed in Section 4.1, there is a trade-off between the final performance and compression ratio. Although the additional quantization layer helps reduce the communication cost, it also causes performance drop. While utilizing the quantization layer, it is important to understand the range of the compression ratio that does not degrade the final performance too much. To this end, we vary the number of clusters $L$ and number of subvectors $q$ in our proposed method and report the resulting accuracy vs. compression trade-off in Figure 4. Note that we define the compression ratio to be the ratio between the original activation size and compressed message (codebook + codewords) size.

One can observe that the proposed method can achieve at least $10\times$ compression ratio with almost no accuracy loss. Furthermore, if we allow for $5\%$ relative loss compared to SPLITFED, then our method can achieve a $490\times$ compression ratio on FEMNIST. This suggests that $99.8\%$ information is redundant and can be dropped during the up-link communication. Surprisingly, on SO Tag, the Recall@5 even improves when adding the quantization layer. We conjecture that compressing the outputs of a intermediate layer might have similar effects to dropout.

**Overall Communication and Computation Efficiencies.** Here, we provide a concrete example showing how the proposed method improves the communication and computation efficiencies over previous works. On FEMNIST, setting $q = 1152$ and $L = 2$ amounts to $490\times$ compression ratio, which amounts to a $490\times$ reduction in the additional communication introduced by the network splitting. Compared to SPLITFED, the overall up-link commutation cost (including the client-side gradients synchronization) is about $10\times$ smaller. Compared to FEDAVG, the up-link communication cost per round is reduced by $62\times$, with $64\times$ less trainable parameters on the clients. We further compare the training curves with respect to total communication costs of FEDLITE against SPLITFED and FEDAVG in Figure 6 in Appendix. Besides, the overall wall-clock time saving may depend on the characteristics of the training system. One can get an estimate using the analytical model in (Agarwal et al., 2021).

**Effectiveness of the Gradient Correction.** From Figure 4, it is easy to observe that the gradient correction technique (i.e., $\lambda > 0$) is crucial for improving performance. Without correction, the algorithm can even diverge in the high compression ratio regime. While $\lambda$ is separately tuned for

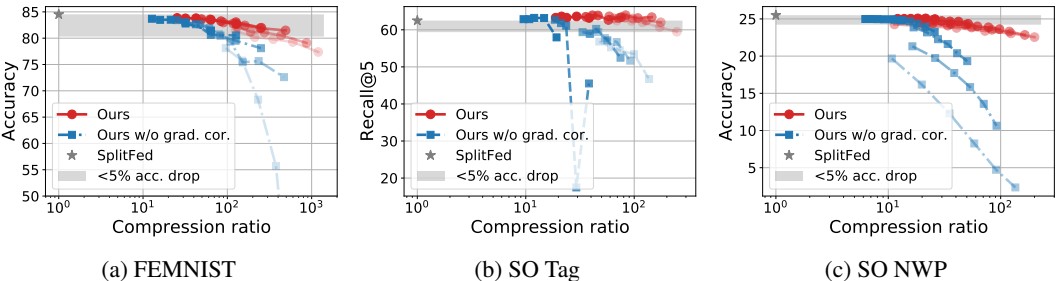

(a) FEMNIST      (b) SO Tag      (c) SO NWP

Figure 4: Trade-off between the accuracy and compression ratio. Our proposed method can achieve up to $490\times$, $247\times$, and $51\times$ communication reduction with less than $5\%$ accuracy drop on FEMNSIT, SO Tag, and SO NWP, respectively. Each curve in the figures corresponds to one value of $q$ (number of subvectors); and each point on a curve corresponds to a specific value of $L$ (number of clusters). For our method, the number of groups $R$ is fixed as $1$.

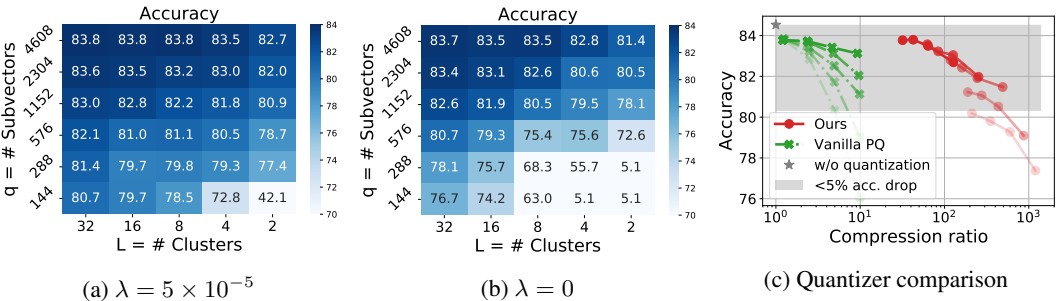

(a) $\lambda = 5 \times 10^{-5}$      (b) $\lambda = 0$      (c) Quantizer comparison

Figure 5: Ablation studies on FEMNIST. (a) and (b): Validation accuracy on FEMNIST when fixing $\lambda$ and varying $q$ and $L$. Setting a small positive value for $\lambda$ improves accuracy for almost all $(q, L)$ pairs; (c) Thanks to subvector grouping, our proposed quantizer can achieve an order of magnitude larger compression ratio as compared to vanilla product quantization scheme.

each point (i.e., each $(q, L)$ pair) in Figure 4, we found that fixing $\lambda$ for all $(q, L)$ pairs still leads to significant improvements. In Figure 5, we report the performance on FEMNIST for various choices of $q$ and $L$. In particular, with $q = 288$, the accuracy improvement ranges from $3 - 72\%$. Furthermore, we found that a small $\lambda$ ranging from $10^{-5}$ to $10^{-3}$ works well on all training tasks in our experiments. Setting a larger $\lambda$ leads to near-zero quantization error. But the model may have poor performance, as it tends to output the same activation for all inputs and becomes meaningless.

**Effectiveness of the Proposed Quantizer.** In Section 4.1, we already show that stacking subvectors into groups (i.e., setting $R < q$) is critical to reach a high compression ratio. But how does it affect the model performance? To answer the question, we run the proposed method with $R = q > 1$ (vanilla PQ), and report the results in Figure 5c. Observe that the proposed method significantly improves the compression ratio with minimal loss of accuracy.

## 6 CONCLUSION

In this paper, we studied the problem of training large machine learning models in federated learning setting with resource-constrained clients. Due to limitations on storage and/or compute capacity, the clients cannot locally optimize the entire model. This prohibits the usage of previous federated learning algorithms. Split neural network is a promising approach to address this issue, as it only assigns the first few (small) layers of a neural network to the clients. However, the network splitting incurs additional communication. In order to make split neural network training to be communication-efficient, we propose an end-to-end training framework FEDLITE that can compress the additional communication by up to $490\times$ with minimal loss of accuracy. The success of our method relies on a variant of product quantization scheme and a gradient correction technique. We perform theoretical analysis as well as extensive experiments on both vision and language tasks to validate its effectiveness.

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

## A    DETAILED EXPLANATION FOR THE EFFECTS OF REGULARIZATION

In Section 4.2, we argue that the client-side gradient is a gradient of the following surrogate loss function

$$s(\boldsymbol{w}_{\mathrm{c}}) = \|\boldsymbol{z}_i - \hat{\boldsymbol{z}}_i\|^2 + \frac{\lambda}{2}\|\boldsymbol{z}_i - \widetilde{\boldsymbol{z}}_i\|^2 \tag{7}$$

where $\hat{\boldsymbol{z}}_i \triangleq \boldsymbol{z}_i - \nabla_{\widetilde{\boldsymbol{z}}_i} h(\boldsymbol{w}_{\mathrm{s}}; \widetilde{\boldsymbol{z}}_i)/2$ and the quantized activation $\widetilde{\boldsymbol{z}}_i$ are fixed vectors that do not have derivatives when computing the gradient. Here, we are going to provide a formal proof to support this argument. Directly taking the derivative, we have

$$\frac{\partial s(\boldsymbol{w}_{\mathrm{c}})}{\partial \boldsymbol{w}_{\mathrm{c}}} = 2(\boldsymbol{z}_i - \hat{\boldsymbol{z}}_i)\frac{\partial \boldsymbol{z}_i}{\partial \boldsymbol{w}_{\mathrm{c}}} + \lambda(\boldsymbol{z}_i - \widetilde{\boldsymbol{z}}_i)\frac{\partial \boldsymbol{z}_i}{\partial \boldsymbol{w}_{\mathrm{c}}} \tag{8}$$

$$= (\nabla_{\widetilde{\boldsymbol{z}}_i} h(\boldsymbol{w}_{\mathrm{s}}; \widetilde{\boldsymbol{z}}_i) + \lambda(\boldsymbol{z}_i - \widetilde{\boldsymbol{z}}_i))\frac{\partial \boldsymbol{z}_i}{\partial \boldsymbol{w}_{\mathrm{c}}}. \tag{9}$$

The above equation is exactly the same as the client-side gradient we defined in (4).

## B    PROOF OF THEOREM 1

### B.1    PRELIMINARIES

Without loss of generality, we define $\boldsymbol{w} = [\boldsymbol{w}_{\mathrm{c}}; \boldsymbol{w}_{\mathrm{s}}]$. Then, the gradient of the original loss function can be decomposed into two parts:

$$\nabla F(\boldsymbol{w}) = [\nabla_{\boldsymbol{w}_{\mathrm{c}}} F(\boldsymbol{w}); \nabla_{\boldsymbol{w}_{\mathrm{s}}} F(\boldsymbol{w})] \tag{10}$$

Similarly, we denote the stochastic gradients as follows:

$$\boldsymbol{g}(\boldsymbol{w}) = [\boldsymbol{g}_c(\boldsymbol{w}); \boldsymbol{g}_s(\boldsymbol{w})] \tag{11}$$

where $\mathbb{E}[\boldsymbol{g}_c(\boldsymbol{w})] = \nabla_{\boldsymbol{w}_{\mathrm{c}}} F(\boldsymbol{w})$ and $\mathbb{E}[\boldsymbol{g}_s(\boldsymbol{w})] = \nabla_{\boldsymbol{w}_{\mathrm{s}}} F(\boldsymbol{w})$. In addition, we assume the stochastic gradients with a mini-batch size of $BS$ have bounded variance:

$$\mathbb{E}\|\boldsymbol{g}(\boldsymbol{w}) - \nabla F(\boldsymbol{w})\|^2 \leq \frac{\sigma^2}{BS}. \tag{12}$$

That is equivalent to

$$\mathbb{E}\|\boldsymbol{g}_c(\boldsymbol{w}) - \nabla_{\boldsymbol{w}_{\mathrm{c}}} F(\boldsymbol{w})\|^2 + \mathbb{E}\|\boldsymbol{g}_s(\boldsymbol{w}) - \nabla_{\boldsymbol{w}_{\mathrm{s}}} F(\boldsymbol{w})\|^2 \leq \frac{\sigma^2}{BS} \tag{13}$$

Then, we denote the client- and server-side gradients in the presence of the quantization layer as follows:

$$\widetilde{\boldsymbol{g}}_c(\boldsymbol{w}) = \boldsymbol{g}_c(\boldsymbol{w}) + \boldsymbol{\delta}_c(\boldsymbol{w}) \tag{14}$$

$$\widetilde{\boldsymbol{g}}_s(\boldsymbol{w}) = \boldsymbol{g}_s(\boldsymbol{w}) + \boldsymbol{\delta}_s(\boldsymbol{w}). \tag{15}$$

The concrete expressions of $\boldsymbol{\delta}_c$ and $\boldsymbol{\delta}_s$ will be derived later.

### B.2    MAIN PROOF

With the above notation, according to the Lipschitz smoothness of the loss function, we have

$$\mathbb{E}[F(\boldsymbol{w}^{(t+1)})] - F(\boldsymbol{w}^{(t)}) \leq -\alpha\left\langle\nabla_{\boldsymbol{w}_{\mathrm{c}}} F(\boldsymbol{w}^{(t)}), \mathbb{E}[\widetilde{\boldsymbol{g}}_c(\boldsymbol{w}^{(t)})]\right\rangle - \alpha\left\langle\nabla_{\boldsymbol{w}_{\mathrm{s}}} F(\boldsymbol{w}^{(t)}), \mathbb{E}[\widetilde{\boldsymbol{g}}_s(\boldsymbol{w}^{(t)})]\right\rangle$$

$$+ \frac{\alpha^2 L}{2}\mathbb{E}\left\|\widetilde{\boldsymbol{g}}_c(\boldsymbol{w}^{(t)})\right\|^2 + \frac{\alpha^2 L}{2}\mathbb{E}\left\|\widetilde{\boldsymbol{g}}_s(\boldsymbol{w}^{(t)})\right\|^2 \tag{16}$$

For the first term on the right-hand side of (16), we have

$$-\left\langle\nabla_{\boldsymbol{w}_{\mathrm{c}}} F(\boldsymbol{w}^{(t)}), \nabla_{\boldsymbol{w}_{\mathrm{c}}} F(\boldsymbol{w}^{(t)}) + \mathbb{E}[\boldsymbol{\delta}_c(\boldsymbol{w}^{(t)})]\right\rangle$$

$$\leq -\left\|\nabla_{\boldsymbol{w}_{\mathrm{c}}} F(\boldsymbol{w}^{(t)})\right\|^2 + \frac{1}{2\epsilon}\left\|\nabla_{\boldsymbol{w}_{\mathrm{c}}} F(\boldsymbol{w}^{(t)})\right\|^2 + \frac{\epsilon}{2}\left\|\mathbb{E}[\boldsymbol{\delta}_c(\boldsymbol{w}^{(t)})]\right\|^2 \tag{17}$$

$$\leq -(1 - \frac{1}{2\epsilon})\left\|\nabla_{\boldsymbol{w}_{\mathrm{c}}} F(\boldsymbol{w}^{(t)})\right\|^2 + \frac{\epsilon}{2}\mathbb{E}\left\|\boldsymbol{\delta}_c(\boldsymbol{w}^{(t)})\right\|^2 \tag{18}$$

$$= -\frac{1}{2}\left\|\nabla_{\boldsymbol{w}_{\mathrm{c}}} F(\boldsymbol{w}^{(t)})\right\|^2 + \frac{1}{2}\mathbb{E}\left\|\boldsymbol{\delta}_c(\boldsymbol{w}^{(t)})\right\|^2 \tag{19}$$

where (17) uses Young's Inequality, and $\epsilon > 0$ is an arbitrary positive number. We set $\epsilon = 1$ right after (17). For the third term on the right-hand side of (16), we have

$$\mathbb{E}\left\|\boldsymbol{g}_c(\boldsymbol{w}^{(t)}) + \boldsymbol{\delta}_c(\boldsymbol{w}^{(t)})\right\|^2 \leq 2\mathbb{E}\left\|\boldsymbol{g}_c(\boldsymbol{w}^{(t)})\right\|^2 + 2\mathbb{E}\left\|\boldsymbol{\delta}_c(\boldsymbol{w}^{(t)})\right\|^2 \tag{20}$$

For the server-side gradients $\widetilde{\boldsymbol{g}}_s$, we can repeat the same process. Combining them together, it follows that

$$\mathbb{E}[F(\boldsymbol{w}^{(t+1)})] - F(\boldsymbol{w}^{(t)}) \leq -\frac{\alpha}{2}\left\|\nabla F(\boldsymbol{w}^{(t)})\right\|^2 + \alpha^2 L\mathbb{E}\left\|\boldsymbol{g}(\boldsymbol{w}^{(t)})\right\|^2$$
$$+ 2\alpha^2 L(\chi_c^2 + \chi_s^2) + \frac{\alpha}{2}(\chi_c^2 + \chi_s^2) \tag{21}$$

where

$$\chi_c^2 = \max \mathbb{E}\left\|\boldsymbol{\delta}_c(\boldsymbol{w})\right\|^2, \tag{22}$$
$$\chi_s^2 = \max \mathbb{E}\left\|\boldsymbol{\delta}_s(\boldsymbol{w})\right\|^2. \tag{23}$$

Using the assumption that the stochastic gradient has bounded variance, we obtain that

$$\mathbb{E}[F(\boldsymbol{w}^{(t+1)})] - F(\boldsymbol{w}^{(t)}) \leq -\alpha(\frac{1}{2} - \alpha L)\left\|\nabla F(\boldsymbol{w}^{(t)})\right\|^2 + \frac{\alpha^2 L\sigma^2}{BS}$$
$$+ 2\alpha^2 L(\chi_c^2 + \chi_s^2) + \frac{\alpha}{2}(\chi_c^2 + \chi_s^2). \tag{24}$$

When $\alpha L \leq 1/4$, we have

$$\mathbb{E}[F(\boldsymbol{w}^{(t+1)})] - F(\boldsymbol{w}^{(t)}) \leq -\frac{\alpha}{4}\left\|\nabla F(\boldsymbol{w}^{(t)})\right\|^2 + \frac{\alpha^2 L\sigma^2}{BS} + 2\alpha^2 L(\chi_c^2 + \chi_s^2)$$
$$+ \frac{\alpha}{2}(\chi_c^2 + \chi_s^2). \tag{25}$$

With minor rearranging and taking the total expectation, it follows that

$$\mathbb{E}\left\|\nabla F(\boldsymbol{w}^{(t)})\right\|^2 \leq \frac{4\mathbb{E}[F(\boldsymbol{w}^{(t)}) - F(\boldsymbol{w}^{(t+1)})]}{\alpha} + \frac{4\alpha L\sigma^2}{BS} + (2 + 4\alpha L)(\chi_c^2 + \chi_s^2). \tag{26}$$

Taking the average from $t = 0$ to $t = T - 1$, obtain that

$$\frac{1}{T}\sum_{t=0}^{T-1}\mathbb{E}\left\|\nabla F(\boldsymbol{w}^{(t)})\right\|^2 \leq \frac{4(F(\boldsymbol{w}^{(0)}) - F(\boldsymbol{w}^*))}{\alpha T} + \frac{4\alpha L\sigma^2}{BS} + (2 + 4\alpha L)(\chi_c^2 + \chi_s^2). \tag{27}$$

If $\alpha = \sqrt{BS/T}$, then

$$\frac{1}{T}\sum_{t=0}^{T-1}\mathbb{E}\left\|\nabla F(\boldsymbol{w}^{(t)})\right\|^2 \leq \frac{4(F(\boldsymbol{w}^{(0)}) - F(\boldsymbol{w}^*))}{\sqrt{BST}} + \frac{4L\sigma^2}{\sqrt{BST}} + \left(\frac{4\sqrt{BS}L}{\sqrt{T}} + 2\right)(\chi_c^2 + \chi_s^2). \tag{28}$$

Next we are going to show how $\chi_c, \chi_s$ relate to the quantization error. Suppose that $\mathcal{B}_i^{(t)}$ denotes the randomly sampled mini-batch on client $i$. Then according to the definition of server-side gradient, we have

$$\widetilde{\boldsymbol{g}}_s(\boldsymbol{w}) = \frac{1}{|\mathcal{S}^{(t)}|}\sum_{i \in \mathcal{S}^{(t)}}\left[\frac{1}{|\mathcal{B}_i^{(t)}|}\sum_{\xi \in \mathcal{B}_i^{(t)}}\frac{\partial h(\boldsymbol{w}_s; \mathcal{Q}(u(\boldsymbol{w}_c; \xi)))}{\partial \boldsymbol{w}_s}\right] \tag{29}$$

$$= \boldsymbol{g}_s(\boldsymbol{w}) + \frac{1}{|\mathcal{S}^{(t)}|}\sum_{i \in \mathcal{S}^{(t)}}\left[\frac{1}{|\mathcal{B}_i^{(t)}|}\sum_{\xi \in \mathcal{B}_i^{(t)}}\left(\frac{\partial h(\boldsymbol{w}_s; \mathcal{Q}(u(\boldsymbol{w}_c; \xi)))}{\partial \boldsymbol{w}_s} - \frac{\partial h(\boldsymbol{w}_s; u(\boldsymbol{w}_c; \xi))}{\partial \boldsymbol{w}_s}\right)\right]. \tag{30}$$

where $\mathcal{Q}$ represents the quantizer and maps the original activation to its quantized version. As a consequence,

$$\boldsymbol{\delta}_s(\boldsymbol{w}) = \frac{1}{|\mathcal{S}^{(t)}|} \sum_{i \in \mathcal{S}^{(t)}} \left[ \frac{1}{|\mathcal{B}_i^{(t)}|} \sum_{\xi \in \mathcal{B}_i^{(t)}} \left( \frac{\partial h(\boldsymbol{w}_\mathrm{s}; \mathcal{Q}(u(\boldsymbol{w}_\mathrm{c}; \xi)))}{\partial \boldsymbol{w}_\mathrm{s}} - \frac{\partial h(\boldsymbol{w}_\mathrm{s}; u(\boldsymbol{w}_\mathrm{c}; \xi))}{\partial \boldsymbol{w}_\mathrm{s}} \right) \right] \tag{31}$$

and

$$\|\boldsymbol{\delta}_s(\boldsymbol{w})\|^2 \le \max_{\xi \in \mathbb{D}_i, i \in \mathcal{I}} \left\| \frac{\partial h(\boldsymbol{w}_\mathrm{s}; \mathcal{Q}(u(\boldsymbol{w}_\mathrm{c}; \xi)))}{\partial \boldsymbol{w}_\mathrm{s}} - \frac{\partial h(\boldsymbol{w}_\mathrm{s}; u(\boldsymbol{w}_\mathrm{c}; \xi))}{\partial \boldsymbol{w}_\mathrm{s}} \right\|^2 . \tag{32}$$

For the ease of writing, we denote $\boldsymbol{z} = u(\boldsymbol{w}_\mathrm{c}; \xi)$ and $\widetilde{\boldsymbol{z}} = \mathcal{Q}(u(\boldsymbol{w}_\mathrm{c}; \xi))$. Using mean value theorem, we have

$$\|\boldsymbol{\delta}_s(\boldsymbol{w})\|^2 \le \max \left\| \frac{\partial^2 h(\boldsymbol{w}_\mathrm{s}; \boldsymbol{u})}{\partial \boldsymbol{u} \partial \boldsymbol{w}_\mathrm{s}} (\widetilde{\boldsymbol{z}} - \boldsymbol{z}) \right\|^2 \tag{33}$$

$$\le \Lambda_1^2 \kappa^2 \tag{34}$$

where $\boldsymbol{u} = t\boldsymbol{z} + (1-t)\widetilde{\boldsymbol{z}}$ for some $0 < t < 1$, constant $\Lambda_1$ is the largest eigenvalue of matrix $\partial^2 h(\boldsymbol{w}_\mathrm{s}; \boldsymbol{u})/\partial \boldsymbol{u} \partial \boldsymbol{w}_\mathrm{s}$, and $\kappa = \max \|\widetilde{\boldsymbol{z}} - \boldsymbol{z}\|$ denotes the maximal quantization error.

Using the same technique, for the client-side gradients, we have

$$\boldsymbol{\delta}_c(\boldsymbol{w}) = \frac{1}{|\mathcal{S}^{(t)}|} \sum_{i \in \mathcal{S}^{(t)}} \left[ \frac{1}{|\mathcal{B}_i^{(t)}|} \sum_{\xi \in \mathcal{B}_i^{(t)}} \left( \frac{\partial h(\boldsymbol{w}_\mathrm{s}; \mathcal{Q}(u(\boldsymbol{w}_\mathrm{c}; \xi)))}{\partial \mathcal{Q}(u(\boldsymbol{w}_\mathrm{c}; \xi))} - \frac{\partial h(\boldsymbol{w}_\mathrm{s}; u(\boldsymbol{w}_\mathrm{c}; \xi))}{\partial u(\boldsymbol{w}_\mathrm{c}; \xi)} \right) \frac{\partial u(\boldsymbol{w}_\mathrm{c}; \xi)}{\partial \boldsymbol{w}_\mathrm{c}} \right]$$
$$+ \frac{1}{|\mathcal{S}^{(t)}|} \sum_{i \in \mathcal{S}^{(t)}} \left[ \frac{1}{|\mathcal{B}_i^{(t)}|} \sum_{\xi \in \mathcal{B}_i^{(t)}} \lambda(u(\boldsymbol{w}_\mathrm{c}; \xi) - \mathcal{Q}(u(\boldsymbol{w}_\mathrm{c}; \xi))) \frac{\partial u(\boldsymbol{w}_\mathrm{c}; \xi)}{\partial \boldsymbol{w}_\mathrm{c}} \right] . \tag{35}$$

Accordingly,

$$\|\boldsymbol{\delta}_s(\boldsymbol{w})\|^2 \le \max \left\| \left( \frac{\partial h(\boldsymbol{w}_\mathrm{s}; \widetilde{\boldsymbol{z}})}{\partial \widetilde{\boldsymbol{z}}} - \frac{\partial h(\boldsymbol{w}_\mathrm{s}; \boldsymbol{z})}{\partial \boldsymbol{z}} + \lambda(\boldsymbol{z} - \widetilde{\boldsymbol{z}}) \right) \frac{\partial \boldsymbol{z}}{\partial \boldsymbol{w}_\mathrm{c}} \right\|^2 \tag{36}$$

$$= \max \left\| \left( \frac{\partial^2 h(\boldsymbol{w}_\mathrm{s}; \boldsymbol{u})}{\partial \boldsymbol{u}^2} - \lambda \right) (\widetilde{\boldsymbol{z}} - \boldsymbol{z}) \frac{\partial \boldsymbol{z}}{\partial \boldsymbol{w}_\mathrm{c}} \right\|^2 \tag{37}$$

$$\le (\Lambda_2 - \lambda)^2 \Lambda_3^2 \kappa^2 \tag{38}$$

where $\Lambda_2$ and $\Lambda_3$ are the largest eigenvalues of matrices $\partial^2 h(\boldsymbol{w}_\mathrm{s}; \boldsymbol{u})/\partial \boldsymbol{u}^2$ and $\partial \boldsymbol{z}/\partial \boldsymbol{w}_\mathrm{c}$, respectively. Substituting the above bounds (34) and (38) on $\boldsymbol{\delta}_c, \boldsymbol{\delta}_s$ into (28), we have

$$\frac{1}{T} \sum_{t=0}^{T-1} \mathbb{E} \left\| \nabla F(\boldsymbol{w}^{(t)}) \right\|^2 \le \frac{4(F(\boldsymbol{w}^{(0)}) - F(\boldsymbol{w}^*))}{\sqrt{BST}} + \frac{4L\sigma^2}{\sqrt{BST}}$$
$$+ \left( \frac{4\sqrt{BS}}{\sqrt{T}} + 2 \right) (\Lambda_1^2 + (\Lambda_2 - \lambda)^2 \Lambda_3^2) \max \|\boldsymbol{z} - \widetilde{\boldsymbol{z}}\|^2 . \tag{39}$$

Here we complete the proof.

## C    More Experimental Details

### C.1    Additional Results

We additionally report the training curves of FedAvg, FedLite and SplitFed on EMNIST in Figure 6. It can be observed that in terms of communication cost, FedLite is significantly faster than other two baselines. We also need to highlight that FedLite and SplitFed have lower memory and computation requirements on clients than FedAvg.

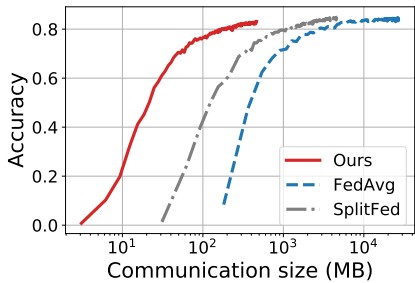

Figure 6: Training curves for different algorithms on FEMNIST. Although split learning-based approaches (SplitFed and ours FedLite) communicate at every iteration, they communicate much less than FedAvg.

## C.2    HYPER-PARAMETER CHOICES

Here, we summarize all the hyper-parameters on three training tasks.

FEMNIST

- Best learning rate: $10^{-1.5}$
- Optimizer: SGD
- Mini-batch size per client $B$: 20
- Activation size $d$: 9216
- Number of clients per iteration: 10
- Ranges of $q$: $\{4608, 2304, 1152, 576, 288, 144\}$
- Ranges of $L$: $\{32, 16, 8, 4, 2\}$
- Ranges of $\lambda$: $\{0, 10^{-5}, 5 \times 10^{-5}, 10^{-4}, 5 \times 10^{-4}\}$
- Client-side model: Same as the first five layers of the neural network used in (Reddi et al., 2020): Conv2d + Conv2d + MaxPool2d + Dropout + Flatten. Model size: $18,816 \times 64$ bits.
- Server-side model: Same as the last three layers of the neural network used in (Reddi et al., 2020): Dense + Dropout + Dense. Model size: $1,187,774 \times 64$ bits.

SO NWP

- Best learning rate: 0.01
- Optimizer: Adam (Kingma & Ba, 2014)
- Mini-batch size per client $B$: 128. Each sample contains 30 words. So the effective batch size is 3840.
- Activation size $d$: 96
- Number of clients per iteration: 50
- Ranges of $q$: $\{48, 24, 12, 6, 3\}$
- Ranges of $L$: $\{960, 480, 240, 120, 60, 30\}$
- Ranges of $\lambda$: $\{0, 5 \times 10^{-4}, 10^{-3}, 5 \times 10^{-3}, 10^{-2}\}$
- Client-side model: Same as the first three layers of the neural network used in (Reddi et al., 2020): Embedding + LSTM + Dense. Model size: $3,680,360 \times 64$ bits.
- Server-side model: Same as the last layer of the neural network used in (Reddi et al., 2020): Dense. Model size: $970,388 \times 64$ bits.

SO Tag

- Best learning rate: $10^{-0.5}$

- Optimizer: AdaGrad (Duchi et al., 2011)
- Mini-batch size per client $B$: 100.
- Activation size $d$: 2000
- Number of clients per iteration: 10
- Ranges of $q$: $\{1000, 500, 250, 200, 125, 25\}$
- Ranges of $L$: $\{100, 60, 40, 20, 10\}$
- Ranges of $\lambda$: $\{0, 10^{-3}, 5 \times 10^{-3}, 10^{-2}, 5 \times 10^{-2}\}$
- Client-side model: One dense layer. Model size: $5000 \times 2000 \times 64$ bits.
- Server-side model: One dense layer. Model size: $2000 \times 1000 \times 64$ bits.

