# OpenReview forum: "FedLite: A Scalable Approach for Federated Learning on Resource-constrained Clients"
_ICLR.cc/2022/Conference — ICLR 2022 Submitted_

### Official Review · Reviewer_oo1D · 2021-10-23

**Correctness:** 3
**Technical Novelty And Significance:** 3
**Empirical Novelty And Significance:** 2
**Recommendation:** 5
**Confidence:** 4

**Main Review:**

**Pros**:
- The paper is well written and structured.
- The paper provides theoretical insight about FedLite convergence rate.
- Interesting and novel idea aim to reduce the communication cost.

**Cons**:
- The authors should cite additional previous works that tried to address the problem of client's device heterogeneity [1,2,3].
- My main concern is that under supervised learning setup FedLite sends the labels to the server to obtain $\nabla_{z_i}h\(w_s;z_i\)$. Sending the labels to the main server compromises privacy (for example in healthcare systems).
- I think the authors should add more challenging experiment to check the trade off between compression and accuracy. (Celeb a from LEAF dataset for example).

**References**
[1] Shamsian, Aviv, Aviv Navon, Ethan Fetaya, and Gal Chechik. "Personalized Federated Learning using Hypernetworks." arXiv preprint arXiv:2103.04628 (2021).
[2] Achituve, Idan, Aviv Shamsian, Aviv Navon, Gal Chechik, and Ethan Fetaya. "Personalized Federated Learning with Gaussian Processes." arXiv preprint arXiv:2106.15482 (2021).
[3] Cho, Yae Jee, Jianyu Wang, Tarun Chiruvolu, and Gauri Joshi. "Personalized Federated Learning for Heterogeneous Clients with Clustered Knowledge Transfer." arXiv preprint arXiv:2109.08119 (2021).

**Summary Of The Paper:**

The authors propose end to end training framework that relies on a novel vector quantization.

The authors claim the following contributions:
- End-to-end communication-efficient neural network splitting-based FL approach for resource-constrained clients.
- New gradient correction scheme for backward pass.
- Reducing communication rounds up to 490X without compromising the accuracy.
- Providing

**Summary Of The Review:**

Questions were raised during the review process.

---

> ### Author Response · Authors · 2021-11-22
> **Response**
>
> We thank the reviewer for the comments! We summarize our responses below.
> 1. **Additional previous works**. Thank the reviewer for the references! Thank you for pointing out the references. If the reviewer feels it's necessary, we can certainly cite those while alluding to the problem of personalized federated learning. However, we would like to state that these papers are not very relevant to the problem that we focus on.
> 2. **Sending labels to the server**. As mentioned in our submission, our proposal (which is aimed at improving the communication efficiency of SplitFed) does not expose any additional information than what is exposed by vanilla split learning. Thus, we can leverage existing privacy preserving schemes that apply to the split learning framework, in general. Towards this, differential privacy (which adds noise to clustered activations) is certainly a viable approach. Another approach is to use some techniques like Insta-Hide, as we mentioned in the introduction. In this scheme, the client data is encrypted before training. To be specific, the training samples are mixed with each other and the pixel signs are randomly flipped.
> 3. **More experiments**. We evaluated our proposed method on three different datasets, covering a variety of training tasks, including image classification, next word prediction, tag prediction. In particular, the StackOverflow dataset contains 342477 total clients, 135818730 total training samples, and more than thousands of classes. In contrast, Celeb in LEAF is a binary classification problem (Smiling vs Not smiling). So we think we already have a larger and more challenging dataset than the Celeb dataset.

---

### Official Review · Reviewer_WW1C · 2021-10-29

**Correctness:** 4
**Technical Novelty And Significance:** 3
**Empirical Novelty And Significance:** Not applicable
**Recommendation:** 5
**Confidence:** 5

**Main Review:**

I truly enjoyed reading this paper. I had thought about the problem before and was excited to read a paper that addresses the communication of split activations in the Split Learning framework. The proposed solution to the problem is presented clearly and the experimental evaluations are thorough.
Regardless, there are a few questions I would like to see answers, as well as some issues I have with the paper, the removing of which would make it a lot stronger.

Introduction & Motivation:
- While Split Learning allows to 'outsource' the final layers' heavy computations to a server, this inherently reduces the privacy compared to FedAvg based approaches that communicate weights-only. Especially since labels have to be communicated for the loss-computation. While I understand that there is a need for SplitLearning regardless of this disadvantage, I would encourage the authors to mention this significant downside to SL in either introduction or discussion.

Background/RW:
- The authors mention model parallel training in some places in their paper. However they do not focus at all on it, provide no experiments for it and ignore questions that arise when using their approach in the model parallel training setting. More specifically, in model-parallel training the additional latency of the quantisation approach cannot be ignored and would presumably cannibalise on any advantages that the reduced communication brings with it. Either the authors include a proper analysis of Model Parallel Training or they should remove the discussion around it.
- There is large body of literature that studies quantised training of neural networks. Some works focus on weight-quantisation, others on activation-quantization (and the combination of both) for the purpose of efficient inference on resource-constrained devices. While many different and sometimes complicated approaches exist, the simplest approaches consider stochastic scalar quantization in combination with the Straight-Through estimator. I am not so familiar with the Split Learning literature, but I would expect this to be a trivial baseline for quantising the activations of a single layer's activations in the forward pass (as considered in this paper). In order to be applicable for sped-up inference in these settings the quantization grid needs to be fixed (optionally learned) at the end of training. In the split learning setting, this grid could be constructed arbitrarily for any chunk of the b x D activation matrix, similarly to the authors' discussion in this paper. Naturally the split-learning approach is not constrained to uniform grids (as would be for inference acceleration on e.g. in8 inputs), so you might even consider quantising a transformation of the activations with e.g. the Normal CDF (assuming activations are normally distributed). I would ask the authors to either consider including experiments for these quantization approaches in their paper or elaborate why this would be out of scope for Split Learning, maybe I am missing something...


Method
- The authors mention that client-weights $w_c$ need to be synchronised every iteration. Is that the default in SplitLearning or would you also consider averaging every E local epochs, as is done in FedAvg traditionally?
- You mention that you assume 'each float-number' to require 64bits. In the context of FedLITE that is relevant for the code-book entries. Is that a requirement for your method to function well? Since NN activations are usually 32bits (16bits are often enough), I am surprised by that assumption.
- Computational complexity: Can you elaborate on how expensive it is to compute the quantised activations for a mini-batch of data? Ideally you would provide timing numbers of forward/backward pass and quantization on a given hardware and discuss the impact on resource-constrained clients.
- You discuss the infeasibility to share codebooks from previous iterations. Did you consider sharing codebooks between data-points of given class, assuming that activations especially higher up in the network share many similarities conditioned on the label? Since you have to transmit the label anyways, decoding would come for free.
- Your discussion surrounding Eq. (5) is not entirely clear to me: What is the benefit from considering Eq (4) as a gradient of the loss in (5)? The first term involving $\hat{z}_i$ is not intuitive to me.
- One of the core insights of Figure 3 seems to be that decreasing R to be =1 seems to have the most favourable scaling properties. Maybe that insight could be reinforced and pointed towards when setting up that hyper-param in the experiment section.
- I did not analyse the provided convergence proof in detail

Experiments
- Hyperparameter selection: How did you end up at the hyper parameters chosen for the base SplitFed algorithm?
- At the last paragraph of 5.1, the authors compare against FedAvg, which assumes a certain number of local epochs E (unless the authors compare against FedSGD). What is the assumption here?
- Does the optimal lambda change significantly with different hyper-parameters L and q? (Figure 4)
- Codebook-design: I would be interested to see the consequences of quantising the codebook to less than $\phi$ bits. Is that something sensible to consider? Additionally, it would be interesting to study the entropy over the selected code-words: Is the method making use of the available code-words? If not, Entropy-coding of the code-words might be a viable approach for further communication reduction.

Small things:
- In the introduction you make a claim about AlexNet, however it sounds like you claim a general CNN architecture.
- Typo above Eq (2): 'while and the remaining layers'
- I understand the need for an acronym, but the letter-selection seems random... :D On the other hand I have no better suggestion.


**Summary Of The Paper:**

The authors propose a method to compress the intermediate activations of a neural network that is split across a client and server in the Federated Learning setting. In this 'Split Learning' scenario, transmitting activations and corresponding gradients incur significant communication costs. The authors discuss FedLITE, a method to compress activations during the forward pass and to correct the gradient to account for the quantization during in the forward pass. Their paper is augmented with an interpretation of the gradient correction term as minimising a surrogate loss function, as well as a convergence analysis of their approach and empirical evaluations.

**Summary Of The Review:**

This is a promising paper and discusses a relevant problem. Some detailed questions remain and some changes I believe are necessary. If the authors address my concerns I will improve my rating of the paper.

---

> ### Author Response · Authors · 2021-11-22
> **Response**
>
> We thank the reviewer for the detailed comments!! It is great to know that the reviewer enjoyed reading this paper. Based on our reading of the reviewer’s comments, we noticed that the reviewer mainly raises two concerns in their review and other comments merely ask for clarifications about our work. We summarize our responses below.
>
> Concerns:
> 1. **Comparison with some naive baselines**. If we understand correctly, the reviewer asked us to compare the proposed scheme with a naive baseline, that is, stochastic scalar quantization. We realized that this baseline is similar to setting the number of subvectors q to be d in product quantization (PQ). In this case, each subvector will become a scalar. As a result, it has the same compression ratio as stochastic scalar quantization. As shown in figure 3, when we increase q to d, the quantization error of PQ will decrease. However, the compression ratio is relatively low. Our proposed subvector grouping strategy and gradient correction strategy still work and can help to significantly improve the compression ratio with minimal accuracy loss.
> 1. **Why choose float-number to be 64 bits?** We chose 64 bits just because we wanted to give a concrete example of what values of the compression ratio can be. As we mentioned in the paper, it can be represented by phi and can be any number. We can also redo all the computation of compression ratios based on 32 bits.
>
> Questions:
> 1. **Do client weights need to be synchronized at each iteration?** Is this default in split learning? Yes, client-side weights need to be synchronized at each iteration and this is the default in previous split learning literature (e.g., see Fig. 1 in [1]).
> 2. **Computation complexity of quantization?** The computation complexity is the same as that of running a K-means algorithm. So the complexity is O(BdLT), where B is the mini-batch size, d is the activation size, L is the number of clusters, T denotes the number of iterations of the clustering algorithm.
> 3. **How about sharing codebooks between data points of a given class?** Do you mean cluster data points based on their labels? If so, we came up with the same idea at the very beginning of this project. We gave up this idea mainly because it is hard to control the compression ratio in this method. In particular, the minimal number of clusters of this method is constrained by the number of classes on the client. Say we have 5 classes in one mini-batch with size 50. Then the maximal compression ratio of this scheme is about 50/5 = 10. However, if we use the proposed method, we can choose the number of clusters to be 2, achieving a compression ratio of 50/2 = 25.
> 4. **How to choose the hyperparameters of SplitFed?** We choose the best one from {10^-2.5, 10^-2, 10^-1.5, 10^-1, 10^-0.5, 10^0, 10^0.5} based on their performance on the validation set.
> 5. **What are the local epochs in FedAvg?** The communication cost is the cost per communication round, so it does not depend on the number of local epochs in FedAvg. In a newly added figure (see Fig. 6), when we compare the training curves of FedAvg and FedLite, we fix the number of local epochs in FedAvg to be 1.
> 6. **Does the optimal lambda change significantly with different hyper-parameters in figure 4?** It does not change significantly. We checked the experimental results and found that nearly all cells take values either 10^-5 or 10^-4.
> 7. **How about quantizing the codebook to less than $\phi$ bits?** We thank the reviewer for pointing out a promising future direction! We agree that there are so many interesting things to explore, as this paper opens up many possibilities.
> 8. **Explanation of Eq. (5)?** We wanted to use this to provide some insights on why the gradient correction technique helps. In equation (5), $\hat{z}_i$ and $\tilde{z}_i$ are fixed vectors which denote the corrected activation and the quantized activation, respectively. Basically, it tells us that the gradient will make the client-side model move in a direction where the activation $z_i$ will be more similar to a linear combination of the corrected activation and the quantized activation.
>
> Suggestions:
> 1. **Mention the need of transferring labels**: Thanks for the suggestion. We now added several sentences in the introduction to reflect this. Whether transferring the labels is needed or it poses some privacy concerns depends on the application setting. In two-party vertical FL [2], the labels are naturally stored on the server. In that case, we do not have the additional privacy issue.
> 2. **Remove the discussion about model parallel training**: We have removed the discussion on model parallel training in the revised version.
> 3. **Emphasize the case R=1**. We have added several sentences in the experiments section regarding this case.
>
> [1] Thapa et al. SplitFed: When federated learning meets split learning
> [2] Romanini et al. 2021. Vertical: A vertical federated learning framework for multi-headed splitnn

---

> > ### Comment · Reviewer_WW1C · 2021-11-23
> > **Thanks for the rebuttal**
> >
> > My major concern about naive baselines is not addressed yet (and mirrored by reviewer ELQm comment 2).
> > Indeed, setting q=d results in scalar quantization. Then there remains a large design space to design the codebook in this setting. K-means in the 1-d space might be a good choice. Assignment of each entry to the closest code-book is deterministic, whereas stochastic quantization in the general case selects the left-lying and right-lying code-entry proportional to its distance to them. Assuming a regular grid, the code-book size can be very small.
> > While I don't doubt that Vector-quantisation can achieve much higher compression ratios, I would like to encourage the authors to show a baseline for stochastic scalar quantization.
> > As for my further comments:
> > - \phi: You haven't answered what bit width you use for your codebook - or should I understand the answer to be 64 bits?
> > - Computational Complexity: You haven't elaborated on empirical measurements for K-means in addition to evaluating the base-model. Since this on-top of forward propagation, in the resource-constrained FL environment, estimating the overhead is important imo.
> > - Grouping / sharing codebooks: Since you argue about R=1 throughout the paper, I wonder why you mention R as part of the design space.
> > - Entropy of codewords: You haven't addressed that question
> > - Local Epochs E in FedAvg: I understand that E doesn't influence communication for a fixed number of rounds, but it does affect the overall performance of FedAvg. A model/dataset trained with E=1 or E=10 or E=20 might perform very differently. Did you tune for this hyperparamer? I am not expecting FedAvg to perform better in the setting of e.g. Femnist and Fig 6, but a fair comparison is warranted.
> > - Finally the too-general claim that relates to AlexNet only remains in your submission.
> >
> > In light of my unaddressed questions, I remain with my rating.

---

### Official Review · Reviewer_ELQm · 2021-11-02

**Correctness:** 3
**Technical Novelty And Significance:** 2
**Empirical Novelty And Significance:** 2
**Recommendation:** 3
**Confidence:** 4

**Main Review:**

The paper proposes to use vector quantization to enhance the communication efficiency of SL, a popular paradigm to conduct federated learning. Both theoretical and empirical results are provided to show the effectiveness.


Pros:
1. The paper is well written, and the research direction of enhancing the communication efficiency of federated learning is potentially impactful.
2. The proposed method is analyzed under both theoretical and empirical lenses.
3. The detailed hyper-parameters and training schedules are provided, which gives good evidence of the reproducibility of the experiments.

Cons:
1. The main concern I have for the current version of the draft is that the proposed FedLite method is not compared against any baseline methods (except for SL itself). Let’s assume we want memory efficient and communication efficient federated learning method, an immediate idea is to use EfficientNet/MobileNet (which are both small and computationally efficient) with FedAvg. How does FedLite look like compared against that vanilla baseline?
2. The paper proposes to use vector quantization without motivation. It is pointed out that the sparsification-based method requires setting the targeted sparsity, but other than the sparsification method, there are still a series of methods to compress tensors, e.g., [1-5]. The authors are expected to at least motivate the proposed method a bit more.
3. I wonder if the communication inefficiency of SL is only caused by the CNN and LSTM architectures. In Transformers, splitting Encoder blocks (which is widely used in pipeline parallelism [6-7]) usually leads to quite good communication efficiency. If so, does it mean FedLite can only attain marginal gains on Transformer based NNs?
4. I’m highly curious about the convergence curve of FedLite compared against FedAvg. It seems FedLite is a FedSGD-ish method, which requires a lot more rounds of communication compared against FedAvg (as stated in the very first FedAvg paper). It is widely known that the communication cost depends on message size and latency (which is related to the round of communication). I wonder if FedLite does lead to significant wall-clock time-saving.
5. One potential approach to reduce the required number of communication rounds for FedSGD-like methods is to increase the batch size. However, since FedLite aims at low-memory training, it limits the space of using large-batch training.
6. A recent technical report indicates that the super-high compression ratio is actually meaningless with reasonable bandwidth [8]. In federated learning applications, the bandwidth is usually limited. Thus, it would be useful to show the actual communication time saving of FedLite.
7. It is stated that FedLite can be combined with existing privacy-preserving methods. But that aspect is not discussed in the paper at all. It’s important for the authors to discuss how to make FedLite private since the vanilla version clearly discloses the users’ privacy.

[1] https://arxiv.org/abs/1802.04434

[2] https://arxiv.org/abs/1610.02132

[3] https://arxiv.org/abs/1905.13727

[4] https://arxiv.org/abs/1806.04090

[5] https://arxiv.org/abs/1705.07878

[6] https://arxiv.org/abs/1806.03377

[7] https://arxiv.org/pdf/2104.04473.pdf

[8] https://arxiv.org/pdf/2103.00543.pdf

**Summary Of The Paper:**

This paper introduces FedLite, a compression method built on top of Split Learning (SL) for better communication efficiency. Under the cases where the size of the intermediate output is large, SL can introduce a communication bottleneck in model training. FedLite tackles that issue via compressing the intermediate output via a vector quantization-based method. A gradient correction scheme is proposed to further enhance the proposed compression method. Convergence analysis and experimental results are provided to justify the effectiveness of FedLite.

**Summary Of The Review:**

Frankly, I think the paper was rushed out. The idea is sweet but clearly requires more effort to show its effectiveness. The current draft only comes with very preliminary experimental results without comprehensive comparisons against various baseline methods. The motivation of the proposed vector quantization method is also not clear. Overall, I do not think the current draft deserves to appear for a presentation at ICLR 2022.

---

> ### Author Response · Authors · 2021-11-22
> **Response**
>
> We thank the reviewer for the detailed comments! Our responses are listed below.
> 1. **Baseline**. We wanted to mention that what the reviewer suggested is not a vanilla baseline. There are mainly two reasons: (1) in order to ensure a fair comparison, when we use EfficientNet or MobileNet in FedAvg, we also need to use them in FedLite. This can help us to further reduce the memory or computation costs; (2) Using small neural networks cannot help us to reduce the size of output space. So when there are a large number of classes (i.e, extremely large classification layer), using small networks in FedAvg does not help.
> 2. **Motivation**. Thank the reviewer for the suggestion! We will expand upon the existing discussion in our paper that tries to motivate our vector quantization scheme. Note that, in the initial version, we mentioned the motivation in several places. For example, in the introduction, we said that “Our proposed solution is based on the critical observation that, given a mini-batch of data, the client does not need to communicate per-example activation vectors if the activation vectors (at the cut layer) for different examples in the mini-batch exhibit enough similarity. Thus, we propose a training framework that performs clustering of the activation vectors and only communicates the cluster centroids to the server”.
> 3. **Communication inefficiency of SL**. We would like to emphasize that the reduction in the communication cost for transmitting the (cut layer) activations to the server realized by our method is independent of the model architecture. Thus, we don’t see why it would lead to smaller gains with Transformer models. Even if the activation size is not very large, the compression ratio of our scheme won’t change.
> 4. **Convergence curve of FedLite and FedAvg**. We now added a new figure 6 in the appendix comparing the convergence curves of FedLite and FedAvg. Even though FedLite communicates after every iteration, it converges faster with respect to the communication cost, as it transfers much less messages per round than FedAvg.
> 5. **Using a large batch size to reduce communication rounds**. Further reducing the communication rounds in FedLite is a very interesting topic. Using a large batch size is only one potential solution with a cost of increased computation and memory on clients (In fact, the available limited resources at the clients can simply prohibit training with very large batches for any algorithm, including FedAvg). One can also extend FedLite to allow multiple local updates at both the server and the clients. This is an interesting yet orthogonal direction to this paper, where we focused on reducing the message size per round.
> 6. **Actual communication time saving**. The actual time saving depends on the system characteristics, such as network delay, communication bandwidth, etc. We have added a reference to [1] that provide ways to compute the actual time savings.
> 7. **Privacy preserving techniques**. As mentioned in our submission, our proposal (which is aimed at improving the communication efficiency of SplitFed) does not expose any additional information than what is exposed by vanilla split learning. Thus, we can leverage existing privacy preserving schemes that apply to the split learning framework, in general. Towards this, differential privacy (which adds noise to clustered activations) is certainly a viable approach. Another approach is to use some techniques like Insta-Hide, as we mentioned in the introduction. In this scheme, the client data is encrypted before training. To be specific, the training samples are mixed with each other and the pixel signs are randomly flipped.
>
> [1] Agarwal et al. 2021. “On the utility of gradient compression in distributed training systems”

---

> > ### Comment · Reviewer_ELQm · 2021-12-05
> > **Thanks for the rebuttal**
> >
> > After carefully reading through the authors' feedback, part of my concerns are addressed. However, my major concern still remains, i.e., it is not clear to me that the proposed FedLite scheme is the only approach to achieve memory-efficient FL. Thus, it is important to understand where FedLite stands compared to other memory-efficient FL schemes. Otherwise, other researchers and engineers will not have a clear idea of when to use FedLite.
> >
> > I thus would like to remain my overall evaluation score.
> >
> > Best,
> > Reviewer ELQm

---

### Official Review · Reviewer_cPmB · 2021-11-02

**Correctness:** 4
**Technical Novelty And Significance:** 2
**Empirical Novelty And Significance:** Not applicable
**Recommendation:** 6
**Confidence:** 4

**Main Review:**

Strengths:
* The approach is well motivated to reduce the transmission cost in split federated learning.
* The gradient correction scheme is a principled solution to tackle the noisy gradients issue and has solid theoretical implications. They also analyzed the theoretical error bound of their proposed approach which further echos the role played by the gradient correction term.
* Their approach shows empirical effectiveness compared with the Split learning baselines. The importance of a positive lambada is clearly analyzed by ablation studies.

Weakness / Questions:
* This paper addresses a specific federated learning scenario, i.e. the split learning setting. However, there are other FL approaches towards efficient parameter transmission or computation cost. I feel that the narrow problem setting that FedLite aims to tackle somehow weakened its contribution. It is unclear to me how FedLite may or may not benefit other FL approaches.
* The experiment section is not extensive enough, as only one splitFed baseline is compared. I am curious to see how the asymptotic performance of the FedLite is dropped compared with non-split learning baselines, especially FedAvg.

**Summary Of The Paper:**

This paper proposed FedLite, which extends a specific federated learning setting called split-learning to be more transmission efficient by a vector quantization scheme. In split federated learning, the classification layer that has a larger parameter set is saved and learned in the server, while the client learns the relatively lightweight feature layers. In FedLite, the activation outputs of the cut layer are compressed by vector quantization before sending to the coordinate server. To alleviate the issue of noisy gradients caused by compressed activations, a regularization term is introduced into the gradient update step which is derived from the Taylor expansion. Empirical results show the effectiveness of the proposed approach, which largely reduces transmission costs without significant performance loss.

**Summary Of The Review:**

This paper is a solid extension of the Split Federated Learning work, which improves transmission efficiency with an acceptable performance drop. The approaches are well motivated and theoretically verified, however the contribution scope of this work is kind of limited and the empirical study part can be further enriched by discussing and comparing with more related work.

---

> ### Author Response · Authors · 2021-11-22
> **Response**
>
> We thank the reviewer for pointing out that our paper is a solid extension of SplitFed and the proposed gradient correction scheme is a principled solution.
>
> Regarding the paper scope, our proposed method can be applied whenever (1) the client resources are too limited to update and store the entire model or the full classification layer; (2) clients need to collaborate in a vertical federated learning setting where the models are naturally split, see [1]. In both settings, which are important and naturally arise in many real-life applications, previous federated learning algorithms are not feasible. So we don’t think the scope of this paper is too narrow.
>
> Regarding the experiments, we additionally report the training curve of FedAvg, FedLite and SplitFed on EMNIST in figure 6. It can be observed that in terms of communication cost, FedLite is significantly faster than other two baselines. We also need to highlight that FedLite and SplitFed have lower memory and computation requirements on clients than FedAvg. We didn’t report the accuracy of FedAvg in the initial submission because the SplitFed paper (see, e.g., Figure 2 and Table 5 in [2])  already compares split learning based approaches with FedAvg, and we show better performance than SplitFed.
>
> [1] Romanini et al. PyVertical: A vertical federated learning framework for multi-headed splitnn.
> [2] Thapa et al. SplitFed: When federated learning meets split learning

---

### Decision · Program_Chairs · 2022-01-20

**Decision:**

Reject

**Comment:**

The paper introduces a compression method for distributed Split Learning for better communication efficiency, by compressing the intermediate output between client and server model by vector quantization. Convergence analysis and experimental results are provided.
Unfortunately consensus among the reviewers remained that it remains slightly below the bar after the discussion phase. Main remaining concerns were the variety of baselines and benefits from split learning setup in experiments, compared to other FL approaches, quantization approaches, architecture splits. Reviewers also missed a discussion of latency requirements of model-parallel training in FL as opposed to data parallel which allows less frequent communication compared to here (e.g. discussing the split layer size vs latency trade-off, here of quantized intermediate layers compared to regular FL). The newly added Figure 6 does not specify or vary the number of local steps (or batch size) in FedAvg.
We hope the detailed feedback helps to strengthen the paper for a future occasion.